# Plasma Assay of Cell-Free Methylated DNA Markers of Colorectal Cancer: A Tumor-Agnostic Approach to Monitor Recurrence and Response to Anticancer Therapies

**DOI:** 10.3390/cancers15245778

**Published:** 2023-12-09

**Authors:** Mojun Zhu, William R. Taylor, Douglas W. Mahoney, Sara S. Then, Calise K. Berger, Kelli N. Burger, Anna M. Gonser, Karen A. Doering, Hao Xie, Patrick H. Foote, Michael W. Kaiser, Hatim T. Allawi, Joleen M. Hubbard, John B. Kisiel

**Affiliations:** 1Department of Oncology, Mayo Clinic, Rochester, MN 55905, USAxie.hao@mayo.edu (H.X.); 2Division of Gastroenterology and Hepatology, Mayo Clinic, Rochester, MN 55905, USAgonser.anna@mayo.edu (A.M.G.); doering.karen@mayo.edu (K.A.D.); foote.patrick@mayo.edu (P.H.F.); 3Division of Clinical Trials and Biostatistics, Mayo Clinic, Rochester, MN 55905, USA; 4Exact Sciences Corporation, Madison, WI 53719, USA; mkaiser@exactsciences.com (M.W.K.);; 5Allina Health Cancer Institute, Minneapolis, MN 55407, USA

**Keywords:** liquid biopsy, neoplasm, residual, DNA methylation, colorectal neoplasms/therapy

## Abstract

**Simple Summary:**

Colorectal cancer (CRC) is the second leading cause of cancer-related death globally. Developing liquid biopsies that aid in the detection and monitoring of recurrent CRC may improve clinical management and patient outcomes. In this study, we investigated a serially sampled blood assay of methylated DNA markers (MDMs) in combination with carcinoembryonic antigen as a clinical tool to surveil and monitor recurrent cancer in a prospective cohort of patients who completed curative-intent therapy for CRC. We demonstrated that the MDMs detected recurrent CRC before the clinical or radiographic detection of recurrence. In a small number of patients, we further showed that the MDMs may correlate with the tumor burden. These data highlight the importance of incorporating epigenetic signatures into liquid biopsies and support further research to validate the correlations of circulating tumor DNA with the tumor burden.

**Abstract:**

Background: Radiographic surveillance of colorectal cancer (CRC) after curative-intent therapy is costly and unreliable. Methylated DNA markers (MDMs) detected primary CRC and metastatic recurrence with high sensitivity and specificity in cross-sectional studies. This study evaluated using serial MDMs to detect recurrence and monitor the treatment response to anti-cancer therapies. Methods: A nested case-control study was drawn from a prospective cohort of patients with CRC who completed curative-intent therapy for CRC of all stages. Plasma MDMs were assayed vis target enrichment long-probe quantitative-amplified signal assays, normalized to *B3GALT6*, and analyzed in combination with serum carcinoembryonic antigen to yield an MDM score. Clinical information, including treatment and radiographic measurements of the tumor burden, were longitudinally collected. Results: Of the 35 patients, 18 had recurrence and 17 had no evidence of disease during the study period. The MDM score was positive in 16 out of 18 patients who recurred and only 2 of the 17 patients without recurrence. The MDM score detected recurrence in 12 patients preceding clinical or radiographic detection of recurrent CRC by a median of 106 days (range 90–232 days). Conclusions: Plasma MDMs can detect recurrent CRC prior to radiographic detection; this tumor-agnostic liquid biopsy approach may assist cancer surveillance and monitoring.

## 1. Introduction

Liquid biopsies, defined in clinical use as the minimally invasive sampling of cells or cellular-derived entities, can be performed on many biological matrices, such as stool, urine, or most commonly, peripheral blood. These have been applied clinically in cancer care to enhance the early detection and surveillance of solid tumor malignancies, guiding intensification and de-escalation of systemic therapy to improve the peri-operative management of locally advanced disease and monitoring the treatment response and identifying resistance to systemic therapy for metastatic disease.

Circulating tumor DNA (ctDNA), as a form of liquid biopsy, aims to detect fragments of nucleic acid chains shed by tumor cells and has been shown to be a promising clinical tool for the detection of molecular residual disease (MRD) in patients with advanced colorectal cancer (CRC). Existing ctDNA testing platforms were reported to detect recurrent CRC with a high positive predictive value (range 85–100%) but limited negative predictive value (range 24–54%) [1,2,3]. Patients with detectable ctDNA after completing definitive treatment were found to have higher rates of recurrent cancer and a worse prognosis [1,2,3,4,5,6,7]. However, there is no conclusive data yet to demonstrate that modifying therapeutic decisions based on ctDNA indeed improves clinical outcomes. It also remains to be determined whether ctDNA can complement and replace radiographic studies and serum carcinoembryonic antigen (CEA) as clinical tools for aiding in the evaluation of the treatment response to anticancer therapies.

The sensitivity and specificity of ctDNA testing as a measure of MRD hinges upon the presence and detection of a subset of genetic and/or epigenetic alterations that drive tumor development when sampled from biological matrices such as plasma. Currently, commercially available ctDNA platforms are either tumor-informed assays that only account for genetic aberrations found in the primary tumor tissue [1] or tumor-agnostic assays that integrate pre-specified panels of genetic and/or epigenetic signatures [3]. These two approaches have not been compared systematically, although there is strong clinical evidence emphasizing the importance of epigenetic alterations, such as DNA methylation, in early tumor pathogenesis and late cancer progression and metastasis [8,9,10,11,12,13,14,15,16]. Our group has identified methylated DNA markers (MDMs) that are strongly associated with de novo CRC and detect recurrent CRC after definitive treatment [17,18]. We previously reported observations suggesting that plasma MDMs might be used as a clinical tool for monitoring the treatment response to anticancer therapies. These attributes specifically include the high concordance of tumor-specific epigenetic alterations as measured by MDMs in tissue between primary and metastatic CRC, minimal epigenetic drift of MDMs in plasma over time and exposure to treatment, and high sensitivity in detecting recurrent CRC [18]. In this study, we sought to further evaluate the potential of circulating MDMs as a novel clinical tool to monitor the treatment response to anticancer therapies in a prospective cohort of adult patients after curative-intent therapy for CRC.

CRC is the third most diagnosed cancer and the third leading cause of cancer-related death in the United States [19]. Despite nationwide promotion of standard CRC screening and the advent of novel systemic therapies, the five-year survival rates for CRC are 65% for all stages but only 14% for those with distant metastasis [19]. Technologies that enable early detection of recurrent cancer and better assessment of the treatment response can help physicians to better plan multimodality anticancer therapies and improve the clinical outcomes of patients with advanced CRC. Although serial liquid biopsies are anticipated to influence the selection of adjuvant therapies, less is known about how ctDNA correlates with the treatment response or tumor burden from micrometastatic disease.

## 2. Materials and Methods

### 2.1. Study Design

This was a nested case-control study drawn from a cohort of adult patients enrolled in a single-center, prospective study that evaluated the plasma assay of candidate MDMs in the setting of surveillance and treatment monitoring of CRC at the Mayo Clinic between 2017 and 2023. The study was conducted in accordance with the Declaration of Helsinki and approved by the Institutional Review Board of the Mayo Clinic (protocol code 17-008184, approved 19 October 2017). All the patients provided signed written informed consent. Consecutive patients with newly diagnosed or recurrent CRC were eligible. Participants underwent serial research blood collections in conjunction with routine cancer surveillance and monitoring tests, which included history and physical examinations, CEA testing, and radiographic imaging studies at regular intervals outlined by the U.S. National Comprehensive Cancer Network guidelines [20]. At baseline, trained study staff recorded: CRC stage and molecular tumor profile at diagnosis; history of tobacco, alcohol, and high-risk familial disorders; and prior cancer diagnoses and their treatment history. At every timepoint of the research blood collection, we abstracted additional clinical data, including the CEA level; treatment exposures to specific anticancer drugs, radiation, and surgery; responses to treatments as radiographic measurements of the tumor burden based on the Response Evaluation Criteria in Solid Tumors version 1.1 (RECIST v1.1) [21]; and clinical assessment of disease status by treating physicians. All the clinical data were verified by an expert GI Medical Oncologist (M.Z.) prior to analysis.

Only patients who received curative-intent, definitive therapy and had at least three post-treatment research blood collections were included in the present study. Patients may have received systemic therapy as part of their curative-intent therapy. For patients with stage I–III disease, the curative-intent therapy included surgical resection of the primary tumor. For patients with stage IV disease, the curative-intent therapy included surgical resection of the primary tumor and definitive treatment of all metastases, which involved either surgical resection or locoregional therapy. We placed the patients into two groups based on whether they developed recurrent CRC (i.e., REC) or they had no evidence of disease (i.e., NED) after completion of curative-intent therapy. Recurrent CRC was defined by radiographic findings that met the criteria for measurable disease based on the RECIST 1.1 or radiographic and/or clinical findings that justified anticancer therapy as determined by the treating physicians.

### 2.2. MDM Assay

A panel of 13 MDMs (*CNNM1*, *ANKRD13B*, *FER1L4*, *ZNF568*, *CHST2*, *ZNF671*, *VAV3*, *QKI*, *GRIN2D*, *DTX1*, *PDGFD*, *SFMBT2*, and *JAM3*) previously shown to have high sensitivity and specificity for CRC in a cross-sectional analysis [18] was assayed for this study. Venous blood was collected in LBgard^®^ tubes (Biomatrica, San Diego, CA, USA) and double-spun and processed at room temperature into platelet-poor plasma by the Biospecimens Accession and Processing facility at the Mayo Clinic, as previously described [22]. Plasma samples were stored at −80 °C. DNA was extracted from 6 mL of plasma (QIAamp Circulating Nucleic Acid Kit, Qiagen, Hilden, Germany) and bisulfite converted (EZ DNA Methylation Kit, Zymo Research, Irvine, CA, USA). The MDMs were assayed via the target enrichment long-probe quantitative-amplified signal (TELQAS, Exact Sciences, Madison, WI, USA) method, as described previously [14]. Using bisulfite-converted DNA, multiplex PCR amplification (12 cycles) of the candidate MDMs was performed prior to 10-fold dilution with Tris/EDTA solution. The MDMs were quantified with *B3GALT6* (reference of bisulfite-converted DNA) in triplex reactions using 10 μL of the diluted amplicons on the ABI 7500DX (Applied Biosystems, Waltham, MA, USA). The Roche Diagnostics (Indianapolis, IN, USA) Cobas e411 Immunoassay Analyzer was used to quantify the CEA.

### 2.3. Statistical Analysis

Continuous patient characteristics were summarized as a median and the 25th–75th percentiles (IQR) and categorical characteristics were summarized as a percent of the subgroup totals. The MDM + CEA panel positivity utilized a previously developed model that had been frozen with a corresponding predefined threshold [18]. The assayed levels of the 13 MDMs and CEA obtained at the sampled timepoints were fed into the model and a score returned (i.e., MDM score). This MDM score was plotted serially over time for each patient along with the threshold for positivity. The estimated concordance-statistic (c-statistic, also known as the area under to curve [AUC]) with the corresponding 95% confidence intervals (CI) for the time to the event endpoints was used to assess the accuracy of the MDM score in predicting future recurrence events [23]. For this assessment, a Cox proportional hazards regression model was fitted by treating the serial measurements of the MDM score as time-dependent covariates [23]. The c-statistics of the MDM score and serially assayed CEA levels alone were compared using a 2-degree of freedom Chi-square test that accounts for the paired nature of the data. The c-statistics between the patient subgroups (unpaired) were based on a Gaussian distribution assumption. A priori, the minimum detectable difference in the paired c-statistics was 0.19, assuming a null AUC of 0.5 and a two-sided significance level of 0.05 at 80% power, with a between assay correlation of 0.7. This calculation also assumed a single measurement per patient and that roughly 50% of the patients would have a positive MDM score. Additional power was gained when the serial measurements were analyzed.

## 3. Results

### 3.1. Patient Characteristics

A total of 36 patients in the prospective cohort met the inclusion criteria for the nested case-control study. One patient was excluded due to the presence of nonspecific, abnormal radiographic findings secondary to pulmonary sarcoidosis that precluded reliable assessment of recurrent CRC based on radiographic studies. Of the included 35 patients, 17 had no evidence of disease (NED) and 18 developed recurrent CRC (REC). The patient and tumor characteristics are well balanced between these two groups (Table 1). With a median follow-up of 1.5 years (IQR 1.4–1.9 years), each patient has at least 3 serial blood samples collected for MDM analysis (median, 5; IQR, 4–6). In the 18 patients who developed recurrent disease, the most common sites of recurrence are the liver (50%), lung (33.3%), and distant lymph nodes (5.6%).

### 3.2. MDM Score Specificity

In the 17 NED patients who were determined not to have recurrent CRC by their treating physicians during the study period, 15 (88%) patients did not have positive MDM scores detected at any timepoint or evidence of recurrent CRC by May 2023, the point of last follow-up for the cohort. Only two patients who were NED during the study time period had positive MDM scores detected at a single timepoint (patients A and B in Figure 1). Both patients went on to develop recurrence after the study period, as described below.

Patient A presented with stage IV cecal adenocarcinoma and two hepatic metastases at the time of diagnosis. He entered the study about 3 weeks after completing curative-intent therapy, which sequentially consisted of 3 months of neoadjuvant chemotherapy with fluorouracil, leucovorin, and oxaliplatin (FOLFOX), followed by combined laparoscopic right colectomy with anastomosis and surgical resection of a liver metastasis, subsequent percutaneous microwave ablation of the other liver metastasis, and 3 months of adjuvant FOLFOX. His first MDM score was positive while his CEA level was mildly elevated at 3.8 ng/mL. Subsequent MDM scores and CEA levels collected at four consecutive timepoints (i.e., 2nd–5th timepoints) over the next 16 months were all negative. This patient was eventually determined to have radiographic recurrence of two new liver metastases 12 months after the last research blood collection (i.e., the 5th timepoint, which was 12 months after the last negative MDM score and 28 months after completion of adjuvant chemotherapy) and underwent successful percutaneous microwave ablation.

Patient B presented with stage III sigmoid colon adenocarcinoma at the time of diagnosis and underwent upfront left hemicolectomy (pT3, pN1a, cM0). She received only 2 months of fluorouracil-based adjuvant chemotherapy due to poor tolerance and entered the study about 12 months after discontinuation of chemotherapy. Following three consecutive negative MDM scores, she was found to have a positive MDM score detected at the 4th timepoint, which was the last research blood collection; this was about 21 months after the last dose of chemotherapy. Interestingly, her MDM scores were consistently rising over time while her CEA levels remained about the same. She had a history of smoking and was followed closely due to the presence of multiple subcentimeter pulmonary nodules. Eventually, she underwent lung wedge resections 19 months after the last research blood collection (i.e., the 4th timepoint, which was 23 months after the last negative MDM score and 41 months after the last dose of chemotherapy). The surgical specimens confirmed metastatic adenocarcinoma in the lung.

Serial negative MDM scores may help to exclude recurrence in the presence of nonspecific radiographic findings. For instance, one patient in the NED group entered the study after presenting with subcentimeter yet new liver lesions on MRI, suspicious for recurrent CRC. About 3 months prior to study enrollment, she underwent surgical resection of a metachronous liver metastasis. Her CEA levels remained within the normal range since initial diagnosis. One of the new liver lesions was biopsied and did not show any evidence of malignancy. Notably, her MDM score at the time of the liver biopsy was negative and remained negative for the next 9 months. Ultimately, this patient was found to have lung recurrence 26 months after the initial liver surgery (15 months after the last negative MDM score). Research blood was not collected at recurrence for this patient.

### 3.3. MDM Score Sensitivity

In the 18 REC patients who were determined to have recurrent CRC by their treating physicians during the study period, the MDM score detected recurrence in 16 (89%). Recurrent CRC in only two patients (C and D in Figure 2a) was not detected by the MDM score, as described below.

Patient C had a history of recurrent, mucinous CRC involving the left ovary and entered the study before proceeding to cytoreductive surgery for a second recurrence, which confirmed peritoneal metastasis with a peritoneal cancer index of 5. Her pre-operative MDM score was negative, and her CEA level was also normal. Subsequently, she had MDM scores and CEA levels collected every 3 months for 15 months and all remained negative and normal. She briefly received chemotherapy for 3 months for a potential third recurrence based on the appearance of two new subcentimeter peritoneal nodules, but this was discontinued after a negative biopsy.

Patient D entered the study after completing adjuvant chemotherapy for a second recurrence involving the periaortic lymph nodes. She had MDM scores and CEA levels collected regularly for 16 months, during which she was found to have a third recurrence involving the retroperitoneal lymph nodes. Additionally, she was found to have new and radiographically enlarging pulmonary nodules and biopsy-proven osseous metastasis at 3 months and 6 months, respectively, after completing curative-intent therapy for the third recurrence. Her disease course indicates persistent micrometastatic disease despite curative-intent interventions. Out of 7 consecutive blood draws, her MDM score was negative once while her CEA level was negative three times. These data are consistent with previous findings that demonstrate the lower sensitivity of liquid biopsy in detecting peritoneal and lymph node disease [18,24].

Serial positive MDM scores may help to detect recurrence preceding notable radiographic findings. For instance, an REC patient entered the study after completing curative-intent surgery for recurrent CRC in the liver. His first MDM score, collected about a month after surgery, was negative. However, the second one, collected 3 months later, was above the positivity threshold. At this time, his surveillance CT scans were reported as no evidence of disease. In a retrospective review, a 1 mm nodule in the left lung was present. Three months later, his third MDM score remained positive, and this left lung nodule grew to 5 mm on the CT scan, which prompted further evaluation. Importantly, his CEA levels stayed normal throughout, and the patient did not report any clinical symptoms that would raise concerns of recurrent cancer. He eventually underwent surgical resection of this lung lesion, which confirmed recurrent CRC.

In addition, we observed that the MDM scores decreased sharply in three patients who received only surgical resection or locoregional therapy for recurrent disease (Figure 2b). Two had liver recurrence and one had a local recurrence at the site of previous colonic anastomosis.

### 3.4. MDM Positive Calls Preceded Clinical or Radiographic Recurrence

In 12 of the 18 REC patients, a positive MDM score was detected preceding clinical or radiographic documentation of recurrent CRC by a median of 106 days (range 90–232 days). In 4 out of 18 patients, a positive MDM score was detected concurrently with clinical documentation of recurrent CRC (Figure 3). Notably, no research blood was collected within 90 days in these 4 patients prior to disease progression.

### 3.5. Accuracy of Predicting Recurrence Based on Serial MDM Score

The c-statistic for the serially measured MDM score was 0.85 (95% CI: 0.75–0.95). There were no statistically significant differences in the MDM score c-statistic when stratified by potentially confounding patient characteristics (Table 2). The c-statistic for the serially collected CEA was 0.76 (95% CI: 0.63–0.86), which was statistically different than that for the MDM score (*p* < 0.0001).

## 4. Discussion

In this study, we showed that circulating MDMs in plasma are highly complementary to CEA and can detect recurrent CRC in patients after receiving definitive treatment preceding radiographic evidence and can discern patients’ treatment response to anticancer therapies. First, we demonstrated that circulating MDMs rise in anticipation of recurrent CRC prior to radiographic detection and remain low in patients without radiographic disease over serial measurements. Second, we showed that serially measured circulating MDMs have the potential to detect changes in the tumor burden after receiving locoregional and/or systemic anticancer therapies.

These findings are supported by other observations reported in the literature. It was shown that ctDNA is much more sensitive than CEA in detecting recurrent CRC [3,4,5]. A postoperative, positive ctDNA in patients who have completed definitive treatment for stage IV CRC was associated with a higher risk of recurrent cancer and shorter survival [1,2,3,4]. Similar observations were also reported in patients who received definitive treatment for stage I–III CRC [3,5,6,7]. Importantly, serial testing [3,5] and incorporating epigenetic signatures [3] in the analysis of ctDNA were shown to further improve the sensitivity of ctDNA to detect recurrence.

In comparison with tumor-informed testing, tumor-agnostic testing has distinct advantages and disadvantages. Tumor-informed testing requires primary or metastatic tissue DNA sequencing to develop a bespoke panel of liquid biopsy targets and further sequencing to exclude clonal hematopoietic events with shared sequence variants. Because tumor-agnostic tests do not require analysis of tumor tissue, the turnaround time and cost of goods and services are anticipated to be more favorable. Historically, tumor-agnostic testing was theorized to have lower performance; these concerns are not supported by current evidence [3,25]. There are few direct comparisons of tumor-informed NGS platform assays to tumor-agnostic PCR platform tests; however, a recent assessment of CRC patient samples found high concordance between the variant allele fraction of a sequencing-based MRD test with the median percent methylation of MDMs using the TELQAS platform. In combining the methods, there were additional cancers detected with minimal specificity trade-off, suggesting that the approaches are complementary [26]. In our study, circulating MDMs in combination with CEA showed a longer lead time to and sensitivity for recurrent CRC than imaging studies and CEA alone in detecting recurrent CRC regardless of the molecular and histological features of the tumor, further supporting the clinical utility of a tumor-agnostic approach.

Our data also add to a growing body of evidence suggesting that ctDNA levels may correlate with the tumor burden and clinical outcomes following locoregional and systemic therapy [25,27]. Mason et al. reported that increased gene mutations (≥4) detected via liquid biopsy were associated with worse 2-year overall survival after resection of colorectal liver metastases [25]. Vidal et al. reported that the percentage change in trunk mutations correlated with the radiographic response to systemic therapy as assessed via the RECIST v1.1 and longer progression-free survival [27]. In line with these findings, we observed decreased MDM scores in three patients following locoregional therapy and directional changes in the MDM scores that correlate with the radiographic response to systemic therapy as assessed vis the RECIST v1.1 in six patients following systemic therapy (Figure 2). These data establish the feasibility of using circulating MDMs to assess the treatment response to multimodality anticancer therapies and support further research in this area.

In addition to the relatively small sample size, this study has several key limitations. First, unmeasurable biases may have been introduced as these data were from an observational cohort of patients who were receiving real-world standard of care clinical therapies and were not randomized to treatment arms. While the positivity threshold for the MDMs was calibrated from data in a prior study [18], this has not been fully validated and the locked 13-gene MDM panel may not be comprehensive. Second, we were not able to assess whether the tumor-agnostic and tumor-informed approaches were complementary. Third, the quantitative MDM score used in this research may not be available from currently available clinical MRD tests, although we propose that this attribute might make them more informative.

## 5. Conclusions

In summary, circulating MDMs in plasma not only detect recurrent CRC but also reflect the treatment response to systemic chemotherapy. Active clinical trials are ongoing to investigate whether ctDNA can better guide adjuvant therapies after surgical resection of CRC [28,29].

## Figures and Tables

**Figure 1 cancers-15-05778-f001:**
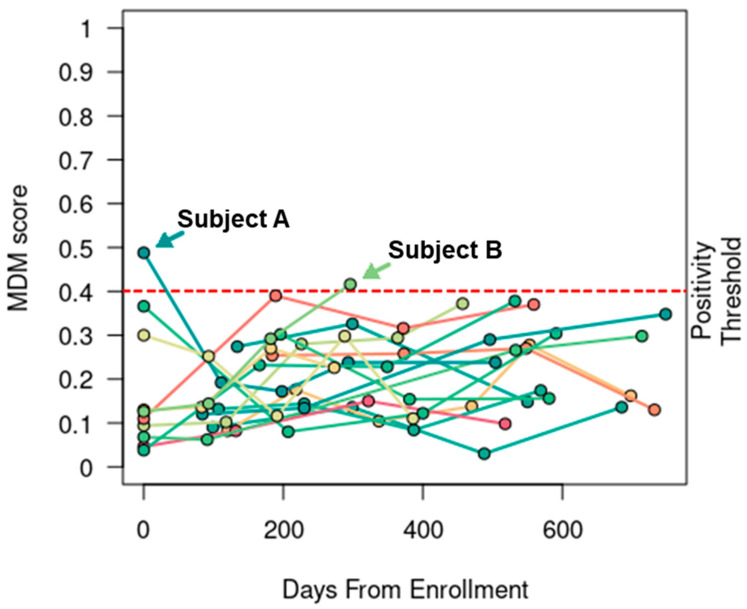
MDM score was specific. In 17 patients without evidence of recurrent CRC, the serial MDM scores were only positive at a single timepoint in two patients (A and B). Both patients eventually developed recurrent disease beyond the study period.

**Figure 2 cancers-15-05778-f002:**
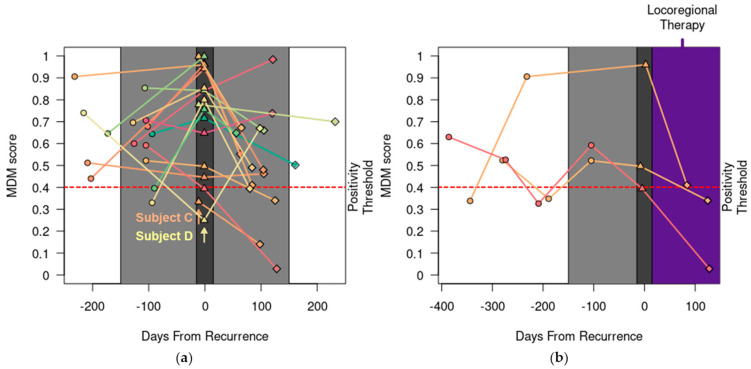
MDM score was sensitive. (**a**) Serial MDM scores detected recurrence (emphasized with dark gray plot shading) in 16 out of 18 patients except for Patient C (peritoneal recurrence) and D (isolated lymph node recurrence). MDM scores pre- (○), post- (◊), and at the time of clinical or radiographic detection of recurrence (∆) are shown. MDM scores 150 days before and after recurrence are emphasized with light gray plot shading. (**b**) Following locoregional therapy for recurrent CRC (shaded in purple), the MDM score decreased in three patients. Dark gray plot shading at the time of recurrence is widened (+/− 15 days) to visually stagger ∆ markers with similar MDM scores.

**Figure 3 cancers-15-05778-f003:**
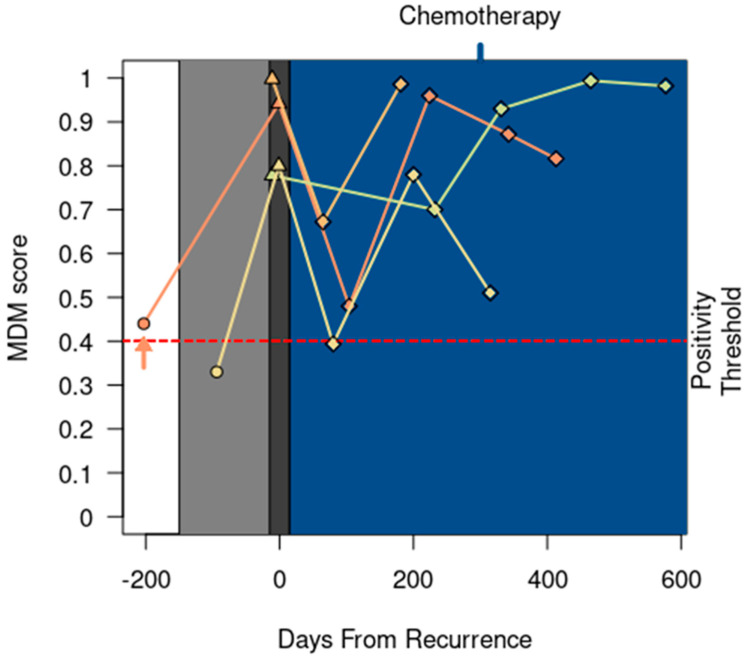
MDM score detected recurrence at the same time as clinical or radiographic detection of recurrent CRC in four patients. One patient (↑) entered the study while receiving chemotherapy for lung recurrences; this patient completed locoregional therapy before starting surveillance. All patients started chemotherapy following clinical recurrence (shaded in blue). MDM scores 150 days before recurrence are emphasized with light gray plot shading. Dark gray plot shading at the time of recurrence is widened (+/− 15 days) to visually stagger ∆ markers with similar MDM scores. Patient status timepoint annotations include surveillance (○), recurrence (∆), and receipt of palliative chemotherapy (◊).

**Table 1 cancers-15-05778-t001:** Patient and tumor characteristics.

		NED (*n* = 17)	REC (*n* = 18)
Age	Median	53	54
	IQR	48–60	48–60
Sex	Female	8 (47.1%)	8 (44.4%)
	Male	9 (52.9%)	10 (55.6%)
Stage at diagnosis	I	0 (0%)	1 (5.6%)
	II	2 (11.8%)	2 (11.1%)
	III	8 (47.1%)	3 (16.7%)
	IV	7 (41.2%)	12 (66.7%)
Tumor sidedness	Left	10 (58.8%)	11 (61.1%)
	Right	7 (41.2%)	7 (38.9%)
*RAS* status	Wild type	5 (29.4%)	10 (55.6%)
	Mutated	5 (29.4%)	8 (44.4%)
	Unknown	7 (41.2%)	0 (0%)
*BRAF* V600E status	Wild type	10 (58.8%)	17 (94.4%)
	Mutated	1 (5.9%)	0 (0%)
	Unknown	6 (35.3%)	1 (5.6%)
Mismatch repair status	Deficient	2 (11.8%)	0 (0%)
	Proficient	15 (88.2%)	18 (100%)

**Table 2 cancers-15-05778-t002:** MDM score area under the curve was not influenced by patient characteristics.

Stratification Variable	Factor Absent	Factor Present	*p* Value
Age ≥ 55	0.78 (0.61–0.94)	0.92 (0.83–1.00)	0.1339
Male sex	0.81 (0.64–0.97)	0.93 (0.85–1.00)	0.2030
Left-sided CRC	0.92 (0.81–1.00)	0.82 (0.68–0.96)	0.2681
Stage IV	0.88 (0.75–1.00)	0.84 (0.7–0.97)	0.6693
Mutant *RAS*	0.77 (0.59–0.96)	0.84 (0.65–1.00)	0.6309

## Data Availability

Data from this research are restricted but may be made available upon written request and with approval from the Mayo Clinic Institutional Review Board, Mayo Clinic Legal and Exact Sciences.

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
