# Peer review of "Plasma Assay of Cell-Free Methylated DNA Markers of Colorectal Cancer: A Tumor-Agnostic Approach to Monitor Recurrence and Response to Anticancer Therapies"

_cancers, 2023, doi:10.3390/cancers15245778_

Round 1
Reviewer 1 Report
Comments and Suggestions for Authors
The manuscript focuses on the evaluation of methylation based approach to evaluate MRD in CRC patients represents a technically correct and timely rleevant manuscript where few minor integratons should be approached to improve the readibility of the manuscript
- In the introduction section, please, could the authors improve clinical meaning of liquid biopsy specimen in this setting?
- In the methodological section, pelase, could the authors improve the description of lcinical data obtained from this analysis?
- In the discussion section, please, could the authors compare NGS and RT-PCR based assays in MRD setting? How these approaches may be integrated?
Comments on the Quality of English Language
Minor english editing is required
Author Response
- In the introduction section, please, could the authors improve clinical meaning of liquid biopsy specimen in this setting?
The opening paragraph of the Introduction is dedicated to describing the current clinical uses of liquid biopsies in cancer (lines 42-57). Lines 58-66 describe the current limitations of this approach. We have added the underlined text to respond to the Reviewer’s request but might need more specifics to better address any ongoing perceived deficits in the context/background on liquid biopsy that already uses the first half of the Introduction to justify the need to have conducted this research.
Liquid biopsies, defined in clinical use as the minimally invasive sampling of cells or cellular-derived entities, can be performed on many biological matrices, such as stool, urine or most commonly, peripheral blood. These have been applied clinically in cancer care to enhance early detection and surveillance of solid tumor malignancies, guiding intensification and de-escalation of systemic therapy to improve peri-operative management of locally advanced disease, and monitoring treatment response and identifying resistance to systemic therapy for metastatic disease.
- In the methodological section, pelase, could the authors improve the description of lcinical data obtained from this analysis?
We appreciate that this section benefits from the inclusion of more detail about participant clinical data and its abstraction. We have added the following (underlined) to the manuscript.
At baseline, trained study staff recorded: CRC stage & molecular tumor profile at diagnosis; history of tobacco, alcohol, and high-risk familial disorders; prior cancer diagnoses and their treatment history. At every timepoint of research blood collection, we abstracted additional clinical data including CEA level, treatment exposures to specific anticancer drugs, radiation and surgery; responses to treatments by as radiographic measurements of tumor burden based on Response Evaluation Criteria in Solid Tumors version 1.1 (RECIST v1.1) [21]; and, clinical assessment of disease status by treating physicians. All clinical data were verified by an expert GI Medical Oncologist (M.Z.) prior to analysis.
If requested, we can provide a copy of our full 6-page data collection form.
- In the discussion section, please, could the authors compare NGS and RT-PCR based assays in MRD setting? How these approaches may be integrated?
We are not aware of very many instances in which these approaches have been compared head-to-head, however we appreciate the opportunity to highlight results from an abstract reported at the American Association for Cancer Research meeting earlier this year, which performed a comparison in 20 CRC patient samples using a tumor informed NGS panel and the TELQAS methylation PCR assay:
There are few direct comparisons of tumor-informed NGS platform assays to tumor-agnostic PCR platform tests; however, a recent assessment of CRC patient samples found high concordance between the variant allele fraction of a sequencing based MRD test with the median percent methylation of MDMs using the TELQAS platform. In combining the methods there were additional cancers detected with minimal specificity trade off, suggesting that the approaches are complementary [26].
Reviewer 2 Report
Comments and Suggestions for Authors
In the study, the authors described an assay of methylated DNA markers (MDMs) in combination with carcinoembryonic antigen to predict recurrence in CRC. The results showed that the MDM score detected recurrence before radiographic evidence.
My recommendations are as follows:
1. Given the small cohort of patients, I recommend reporting CEA positivity in the recurrent and disease-free cases to show the improved sensitivity and specificity brought by the MDM score.
2. It would be interesting to see if certain MDM would detect specific recurrence sites (e.g. liver) in the MDM panel.
Author Response
- Given the small cohort of patients, I recommend reporting CEA positivity in the recurrent and disease-free cases to show the improved sensitivity and specificity brought by the MDM score.
We are grateful for this recommendation. As currently written, the C-statistics (area under curve) were calculated for the predictive accuracy of both MDMs+CEA and CEA alone for predicting future recurrence events. This analysis takes into account the serial measurements of the classifiers as time to event co-variates (lines 144-153). Thus we are already reporting the predictive value of cumulative measurements to estimate recurrence in the next 3 months and showing a highly significant gain when including MDMs over CEA alone (lines 280-283). Because serial testing is how these tests are (or would be) used in clinical practice, our current reporting provides more information than an estimate of sensitivity and specificity from a single, arbitrarily chosen, point in time.
- It would be interesting to see if certain MDM would detect specific recurrence sites (e.g. liver) in the MDM panel.
We appreciate this interesting question from the Reviewer. Based on the methodology by which the MDMs were selected, we anticipate that the MDM profile will not be site of metastatic organ specific. In particular, the markers were chosen for their similarity between the primary tumor and either synchronous or metachronous metastatic disease (Ref 18. Xie, et al.). In a previous publication, we did report that MDM sensitivity appeared to be lower in organ specific metastatic sites, such as lymph nodes or peritoneal nodules and higher in liver and lung metastases. We have added that reference to line 242 which describes the finding of variable/decreased sensitivity for peritoneal and lymph node metastasis.